# Association of Objectively Measured Physical Activity with Physical Function in Patients with Sarcopenia during Hospitalized Rehabilitation

**DOI:** 10.3390/nu14204439

**Published:** 2022-10-21

**Authors:** Takuro Ohtsubo, Masafumi Nozoe, Masashi Kanai, Katsuhiro Ueno, Mai Nakayama

**Affiliations:** 1Department of Rehabilitation, Nishi-Kinen Port Island Rehabilitation Hospital, Kobe 650-0046, Japan; 2Department of Physical Therapy, Faculty of Nursing and Rehabilitation, Konan Women’s University, Kobe 658-0001, Japan

**Keywords:** physical activity, sarcopenia, hospitalized rehabilitation, activity of daily living

## Abstract

This study aimed to investigate the association between objectively measured physical activity and functional improvement in hospitalized patients with sarcopenia. In this retrospective cohort study, physical activity (light-intensity physical activity [LIPA]; moderate-to-physical activity [MVPA]) was measured using a triaxial accelerometer in patients with sarcopenia undergoing rehabilitation on hospital admission. The primary outcome was physical function measured with the SPPB and activity of daily living (ADL) measured with the functional independence measure scores for motor function (FIM-M) at hospital discharge. Multiple regression analysis was per-formed to investigate the relationship between the objectively measured physical activity and functional outcomes. A total of 182 patients with sarcopenia (aged 81; interquartile range (IQR) 13 years) were included in this study. In the multiple regression analysis, LIPA was associated with the SPPB score at discharge (β = 0.180, *p* = 0.015) but not with FIM-M at discharge. MVPA was not associated with SPPB or FIM-M scores at discharge. In conclusion, LIPA on admission is independently associated with physical function, but not ADL, in patients with sarcopenia undergoing hospitalized rehabilitation.

## 1. Introduction

Sarcopenia is a disease characterized by the loss of muscle mass and strength or physical function [1,2] and is associated with clinical adverse outcomes such as falling and fracture [3], functional decline [4], and mortality [5,6]. Among hospitalized rehabilitation patients, sarcopenia has a higher prevalence [7] and is associated with worse recovery of ability to complete activities of daily living (ADL) [8,9,10]. Thus, the diagnosis of sarcopenia and its treatment in rehabilitation hospitals is important [11,12].

In a recent meta-analysis, physical exercise, protein or nutrition supplementation, and aerobic exercise were the most effective interventions to improve physical performance in sarcopenia in any setting [13]. Another systematic review reported that multi-component physical exercise intervention provided the strongest results in preventing sarcopenia in hospitalized older adults [14]. Furthermore, another systematic review suggested that exercise and nutrition interventions may be useful in improving physical function in patients with sarcopenia [15]. Therefore, management of physical exercise and nutrition in patients undergoing rehabilitation is important [16].

Not only physical exercise but also physical activity which includes physical exercise or ADL has a positive impact on muscle mass and function and improves physical performance in older adults [17]. Particularly in community-dwelling older adults, increasing moderate-to-vigorous-intensity physical activity (MVPA) improves physical function of sarcopenia [18,19,20]. However, the relationship between the amount of physical activity and functional recovery in patients with sarcopenia during hospitalized rehabilitation has not been studied, except for the results of a cross-sectional study [21]. This may contribute to the development of effective interventions by investigating the associations among hospitalized rehabilitation patients.

This study aimed to investigate the association between objectively measured physical activity and functional improvement in patients with sarcopenia during hospitalized rehabilitation.

## 2. Materials and Methods

### 2.1. Participants

This retrospective cohort study included patients aged ≥20 years who were transferred to a 100-bed rehabilitation hospital from another acute care hospital once they were medically stable between January 2020 and April 2022. The primary diseases they suffered from were musculoskeletal disorders (e.g., hip fracture, vertebral compression fracture, hip or knee arthroplasty), neurological diseases (e.g., stroke, traumatic brain injury, and spinal cord disease), and other diseases (e.g., infections, cardiovascular diseases). We excluded patients who had premorbid-dependent gait and could not perform measurements because of pacemaker implantation, limb defects, orthopedic treatment devices, oedema and altered hydration states, and lack of cooperation for measurements due to consciousness disorder or cognitive dysfunction. After the diagnosis of sarcopenia based on the Asia Working Group for Sarcopenia (AWGS) 2019 criteria, non-sarcopenic patients and patients with higher physical function (Short Physical Performance Battery; SPPB >9) were also excluded.

All rehabilitation programs were designed to improve ADLs according to the patient’s functional disability and ability impairment. The amount of rehabilitation was more than 2 h and less than 3 h per day. These programs included joint range of motion, muscle strengthening, standing, walking, swallowing, cognitive, and ADLs training. Nutritional management was designed by registered dietitians and a nutrition support team according to each patient’s condition.

This study was conducted with the approval of the Institutional Ethics Committee, and we provided an opt-out option to allow patients to withdraw their participation from this study.

### 2.2. Assessment and Data Collecting

Demographic and general medical history data were collected upon admission, including age, sex, primary reason for admission (musculoskeletal, neurological, or other disorders), length of acute hospital stay (LOS), Charlson comorbidity index (CCI) [22], and body mass index (BMI). We also estimated appendicular skeletal muscle mass measured using bioelectrical impedance analysis (BIA) and measured functional independence measure (FIM) scores for motor function (FIM-M) and cognitive function (FIM-C) on admission [23]. Nutritional status was assessed using the Mini Nutritional Assessment-Short Form (MNA^®^-SF) [24], muscle strength was assessed using hand grip strength (HG), physical performance was assessed using SPPB and gait speed. If the participants were walking independently, they were instructed to walk along a 10-m walkway at a comfortable gait speed. Walking speed was calculated using walking time.

### 2.3. Diagnosis of Sarcopenia

Sarcopenia was diagnosed based on AWGS 2019 criteria using muscle mass, muscle strength, and physical performance [2]. At admission and 3 h after lunch, muscle mass was measured using bioelectrical impedance analysis (InBody S10, InBody) [9,21] in the supine position. Appendicular muscle mass was estimated from the appendicular lean mass obtained by measuring BIA, and SMI was calculated as the measured appendicular muscle mass divided by the height squared (m^2^). The cut-off values for SMI were <7.0 and <5.7 kg/m^2^ for men and women, respectively [2]. Handgrip strength was measured as muscle strength using a digital grip strength meter (Grip-D T.K.K.5401, Takei Kiki Kogyo) with both hands in a standing or seated position, with arms straight at their side. The highest value of the three measurements was recorded as muscle strength. In patients with hemiparesis, it was measured using the non-paralyzed hand. Low muscle strength was defined as <28 kg for men and <18 kg for women [2]. Low physical performance was defined as a Short Physical Performance Battery (SPPB) score ≤9 or gait speed <1.0 m/s [2]. Sarcopenia was diagnosed in patients with both low muscle mass and low muscle strength, or both low muscle mass and low physical performance, in accordance with the AWGS2019 criteria [2].

### 2.4. Physical Activity Measurements

Physical activity was measured using an activity monitor with a triaxial accelerometer (Active Style Pro HJA750-C; OMRON, Kyoto, Japan). Activity monitoring provides highly accurate metabolic equivalent (MET) estimations for a wide range of body motions in ADL and discriminates between ambulances and other activities [25,26]. Reliability and validity of sitting, standing, and walking activities have been demonstrated in patients with subacute stroke [27,28]. All participants wore the activity monitor attached to their belts between 7:00 AM and 7:00 PM, which included the rehabilitation time, except during bathing, for 5 consecutive days [21,29]. MET data were recorded every 60 s for 12 h and processed using the manufacturer’s software (HMS-HJA-IC01J; OMRON, Kyoto, Japan). The MET data were classified into two categories based on intensity: LIPA was 1.6–2.9 METs, and MVPA ≥3 METs [21,29]. The total minutes and average duration of LIPA and MVPA were calculated daily for 5 consecutive days [21,29].

### 2.5. Outcomes

The primary outcomes were the Short Physical Performance Battery (SPPB) and FIM-M scores at discharge from the convalescent rehabilitation ward. The SPPB consists of three components: walking speed, the chair-stand test, and standing balance. Each component was scored from 0 (not possible) to 4 (best performance); the scores add up to a total score ranging from 0 (worst performance) to 12 (best performance) [30]. The FIM consists of 18 items ranging from total assistance to complete independence, with each item on a 7-point ordinal scale. The FIM-M consists of 13 items from the motor category including self-care [23]. Both assessments were performed by physical or occupational therapists at the time of discharge.

### 2.6. Sample Size

The sample size was calculated using G*Power version 3.1.9.2 (Hein-rich-Heine-Universität Düsseldorf, Düsseldorf, Germany). A total of 127 older rehabilitation patients (effect size, 0.15; alpha error probability, 0.05; power, 0.8; number of predictors, 12) were included in the regression analysis.

### 2.7. Statistical Analysis

The results are reported as median and interquartile ranges and as percentages (%) for categorical data. For the multiple regression analysis, the dependent variables were the SPPB and FIM-M scores at discharge, even after adjusting for age, sex, BMI, the primary reason for admission, length of rehabilitation hospital stay, CCI, MNA-SF, LIPA, MVPA, FIM-C at admission, and FIM-M at admission (dependent variable; FIM-M) or SPPB at admission (dependent variable; SPPB). The variance inflation factor (VIF) was measured to define multicollinearity; a VIF value between 1 and 10 indicated a lack of multicollinearity. All statistical analyses were performed using IBM SPSS Statistics, version 22 (IBM Corp., Armonk, NY, USA). Statistical significance was set at *p* < 0.05.

## 3. Results

A total of 859 rehabilitation patients were hospitalized during the study period. Among them, 160 patients were excluded because they had premorbid-dependent gait (*n* = 40), pacemaker implantation (*n* = 12), limb defects and orthopedic treatment devices (*n* = 11), oedema and altered hydration states (*n* = 14), and consciousness disorder or cognitive dysfunction (*n* = 83). Thus, 699 patients met the inclusion criteria. Among them, 517 patients were excluded from the analysis due to a lack of informed consent (*n* = 17), lack of outcome measures (*n* = 4), transfer to another hospital (*n* = 18), failure to discharge from the hospital during the study period (*n* = 2), death during hospitalization (*n* = 1), loss of accelerometers or difficulty in wearing them continuously (*n* = 28), failure of the measurement device (*n* = 65), non-sarcopenic patients (*n* = 257), good physical function among patients with sarcopenia (*n* = 43), and missing data (*n* = 82). A total of 182 patients were enrolled in this study.

Table 1 shows the baseline characteristics at admission and the rehabilitation out-comes for all participants. The median patient age was 81 years (IQR 13 years). Among these patients, 66% were female. The primary reasons for admission were neurological disorders (*n* = 87, 48%), musculoskeletal disorders (*n* = 86, 47%), and other diseases (*n* = 9, 5%). The median SPPB score was 4 (7), and the FIM-M score was 39 (26) at admission. The median LIPA was 86 (83) min, and MVPA was 2 (3) min. At discharge, the SPPB score was 8 (9), and the FIM-M score was 78 (34). The median length of rehabilitation hospital stay was 88 (77) days, and 128 (70) patients were discharged home.

Table 2 and Table 3 show the results of the multiple regression analysis with the SPPB score or FIM-M score at discharge as the dependent variables. LIPA was associated with SPPB score at discharge (β = 0.180, *p* = 0.015) but not with FIM-M at discharge (β = 0.100, *p* = 0.196). MVPA was not associated with the SPPB score (β = −0.078, *p* = 0.212) or FIM-M (β = 0.039, *p* = 0.524) at discharge.

## 4. Discussion

This retrospective cohort study investigated the association between objectively measured physical activity and functional improvement in patients with sarcopenia during hospitalized rehabilitation. We showed that the amount of LIPA on admission was associated with physical function but not with ADL ability at discharge in this population. To our knowledge, this is the first study to show an association between physical activity and functional outcomes in patients with sarcopenia.

Previous studies have shown that in community-dwelling older adults, increasing MVPA improves physical function and relieving sarcopenia symptoms [18,19,20]. In our study, LIPA on admission was associated with physical function at discharge, but MVPA on admission was not.

One possible reason for this result is the difference in physical functioning between community-dwelling and inpatients [16,31]. Participants who were hospitalized for rehabilitation often had decreased physical function, and we also excluded patients with higher physical function (i.e., SPPB ≥ 10). Thus, our subjects did not perform higher-intensity exercises or activities included in MVPA. Based on this result, it may be important to increase LIPA to improve physical function in patients with sarcopenia during rehabilitation hospitalization, even though it is difficult to perform MVPA.

Some studies have reported a positive association between physical activity and ADL or the incidence of disability in older adults in several settings [32,33,34]. However, our results suggest that physical activity is not associated with ADL. This discrepancy may also be due to the lower physical function of our participants than that in other studies. On the other hand, nutritional status on admission was negatively associated with ADL at discharge, which is similar to previous studies [35,36]. This result implies the importance of nutrition assessment and management among inpatients with sarcopenia during hospitalization to improve their ADL.

This study had some limitations. First, this study was conducted at a single rehabilitation hospital, making it difficult to generalize the results. Second, we could not include patients with pacemakers or other electronic implants, amputations, oedema, or altered hydration states. Third, physical activity was measured only early after admission; thus, it was impossible to determine whether the increase in LIPA or MVPA was related to physical functional improvement. Fourth, our participants included four patients who were suffering from COVID-19 infection. The previous study showed a negative association between this infection and physical function [37]. However, because there were very few of them, we could not investigate the effect of COVID-19 infection on physical function. Finally, the amount of physical activity measured in this study was included during the rehabilitation period. Rehabilitation might affect the amount and intensity of activity through the intervention method, which may affect physical activity. These methodological limitations may have affected the physical activity results of this study. Future research should investigate the effects of change in physical activity on functional outcomes in this population.

## 5. Conclusions

LIPA on admission is independently associated with physical function, but not ADL, in patients with sarcopenia undergoing rehabilitation hospitalization. It may be important to increase LIPA levels to improve physical function in patients with sarcopenia during hospitalized rehabilitation.

## Figures and Tables

**Table 1 nutrients-14-04439-t001:** Baseline characteristics at admission and the rehabilitation outcomes for all participants.

	Total Cohort (*n* = 183)
Age, years, median (IQR)	81 (72, 85)
Sex, Female, n (%)	120 (66)
Primary reason for admission, n (%)	
Neurological	87 (48)
Musculoskeletal	86 (47)
Other	9 (5)
Length of acute hospital stay, days, median (IQR)	24 (17, 31)
Charlson Comorbidity Index, score, median (IQR)	2 (0, 3)
Body mass index, kg/m^2^, median (IQR)	21 (19, 23)
MNA^®^-SF, score, median (IQR)	7 (5, 8)
MNA^®^-SF, n (%)	
Normal	1 (1)
At risk	73 (40)
Malnutrition	108 (59)
FIM on admission, score, median (IQR)	
Motor	39 (26, 52)
Cognitive	21 (13, 29)
SPPB, score, median (IQR)	4 (0, 7)
Gait speed, m/sec, median (IQR)	0.70 (0.49, 0.90)
Physical activity, min, median (IQR)	
LIPA	86 (48, 131)
MVPA	2 (1, 4)
Length of rehabilitation hospital stay, days, median (IQR)	88 (60, 137)
Length of total hospital stay, days, median (IQR)	109 (84, 173)
FIM at discharge, score, median (IQR)	
Motor	78 (52, 86)
Cognitive	33 (21, 35)
SPPB at discharge, score, median (IQR)	8 (2, 11)
Functional improvement (SPPB > 9 at discharge), n (%)	118 (65)
Gait speed at discharge, m/sec, median (IQR)	0.90 (0.60, 1.06)
Home discharge, n (%)	128 (70)

IQR, interquartile range; MNA^®^-SF, mini nutritional assessment-short form; FIM, functional independence measure; SPPB, short physical performance battery; LIPA, light-intensity physical activity; MVPA, moderate-to-vigorous physical activity.

**Table 2 nutrients-14-04439-t002:** Multivariate linear regression analysis for SPPB score at discharge.

	β	*p*-Value
Age	−0.083	0.177
Sex		
Male	Reference	-
Female	0.055	0.384
Body mass index	−0.052	0.443
Primary reason for admission		
Neurological	Reference	-
Musculoskeletal	0.002	0.975
Other	−0.089	0.141
Length of rehabilitation hospital stay	−0.168	0.023
Charlson Comorbidity Index	−0.039	0.547
MNA-SF	0.129	0.085
SPPB on admission	0.426	<0.001
FIM-Cognition on admission	0.051	0.437
LIPA	0.180	0.015
MVPA	−0.078	0.212
	Adjusted R^2^	0.438

MNA^®^-SF, Mini Nutritional Assessment-Short Form; FIM, Functional Independence Measure; SPPB, Short Physical Performance Battery; LIPA, light-intensity physical activity; MVPA, moderate-to-vigorous physical activity.

**Table 3 nutrients-14-04439-t003:** Multivariate linear regression analysis for FIM-Motor score at discharge.

	β	*p*-Value
Age	−0.045	0.443
Sex		
Male	Reference	-
Female	0.026	0.667
Body mass index	−0.029	0.660
Primary reason for admission		
Neurological	Reference	-
Musculoskeletal	0.068	0.358
Other	0.022	0.712
Length of rehabilitation hospital stay	−0.136	0.059
Charlson Comorbidity Index	−0.039	0.543
MNA-SF	0.167	0.023
FIM-Motor on admission	0.274	0.002
FIM-Cognition on admission	0.192	0.006
LIPA	0.100	0.196
MVPA	0.039	0.524
	Adjusted R^2^	0.470

MNA^®^-SF, Mini Nutritional Assessment-Short Form; FIM, Functional Independence Measure; SPPB, Short Physical Performance Battery; LIPA, light-intensity physical activity; MVPA, moderate-to-vigorous physical activity.

## Data Availability

The data presented in this study are available upon request from the corresponding author following permission by the Ethics Committee and the hospital at which the study was conducted.

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
