# Peer review of "Association of Objectively Measured Physical Activity with Physical Function in Patients with Sarcopenia during Hospitalized Rehabilitation"

_nutrients, 2022, doi:10.3390/nu14204439_

Round 1

Reviewer 1 Report

The study by Takuro Ohtsubo et al is about the effect of light exercise (LIPA) or moderate exercise (MVPA) on patients hospitalized for various causes but all having sarcopenia measured by the Asia Working Group for Sarcopenia (AWGS) 2019 criteria.

The study design is interesting because both the physical enhancement programs, including dietary, and the procedures used for the various measures are objective and adequately reproducible.

The limitations of the scientific work performed are also discussed by the authors very accurately and without undue emphasis.

The sample selection is statistically correct and the inclusion and/or exclusion criteria rigorous.

The results obtained are original and interesting, especially with regard to their potential dissemination in other rehabilitation facilities interested in the treatment of sarcopenic status.

Given the current situation I would suggest to the authors, if they have available data concerning possible COVID-19/Omicron infections for the selected patients, to include them in the Table in which they describe the characteristics of the statistical sample. This is because the effects of the pandemic include the presence of a syndrome defined as long-covid whose effects are directly related to physical capacity.

Author Response

Point 1:

The study by Takuro Ohtsubo et al is about the effect of light exercise (LIPA) or moderate exercise (MVPA) on patients hospitalized for various causes but all having sarcopenia measured by the Asia Working Group for Sarcopenia (AWGS) 2019 criteria.

The study design is interesting because both the physical enhancement programs, including dietary, and the procedures used for the various measures are objective and adequately reproducible.

The limitations of the scientific work performed are also discussed by the authors very accurately and without undue emphasis.

The sample selection is statistically correct and the inclusion and/or exclusion criteria rigorous.

The results obtained are original and interesting, especially with regard to their potential dissemination in other rehabilitation facilities interested in the treatment of sarcopenic status.

Given the current situation I would suggest to the authors, if they have available data concerning possible COVID-19/Omicron infections for the selected patients, to include them in the Table in which they describe the characteristics of the statistical sample. This is because the effects of the pandemic include the presence of a syndrome defined as long-covid whose effects are directly related to physical capacity.

Response 1: First, we would like to thank you for recognizing the significance of our research. And we also thank you for your comments that will lead us to improve our manuscript.

It was known that the effects of COVID-19 infection on poor physical function as you showed, but we have only four patients with COVID-19. Thus, we could not investigate the effects of COVID-19 on physical function because of the small sample.  We added these explanations in the discussion section following your comment.

Fourth, our participants include four patients who were suffering from COVID-19 in-fection. The previous study showed a negative association between this infection and physical function [38]. However, there were very few of them, we could not investigate the effect of COVID-19 infection on physical function.

[38] Belli, S.; Balbi, B.; Prince, I.; Cattaneo, D.; Masocco, F.; Zaccaria, S.; Bertalli, L.; Cattini, F.; Lomazzo, A.; Dal Negro, F.; Giardini, M.; Franssen, F.; Janssen, D.; Spruit, M. A. Low physical functioning and impaired performance of activities of daily life in COVID-19 patients who survived hospitalisation. The European respiratory journal. 2020, 56, 2002096; DOI: 10.1183/13993003.02096-2020.

Reviewer 2 Report

The paper entitled “Association of objectively measured physical activity with physical function in patients with sarcopenia during hospitalized rehabilitation” is beneficial for the field of patient rehabilitation. The paper is well written with the appropriate structure without serious spelling errors. The research methods are appropriately chosen, and the results are comprehensible and statistically evaluated in an appropriate manner. The aim of the study to investigate the association between objectively measured physical activity and functional improvement in patients with sarcopenia during hospitalization was met. I recommend accepting this paper for publication in the journal Nutrients.

Author Response

Point 1:

The paper entitled “Association of objectively measured physical activity with physical function in patients with sarcopenia during hospitalized rehabilitation” is beneficial for the field of patient rehabilitation. The paper is well written with the appropriate structure without serious spelling errors. The research methods are appropriately chosen, and the results are comprehensible and statistically evaluated in an appropriate manner. The aim of the study to investigate the association between objectively measured physical activity and functional improvement in patients with sarcopenia during hospitalization was met. I recommend accepting this paper for publication in the journal Nutrients.

Response 1: First, we would like to thank you for recognizing the significance of our research. And we also thank you for your comments that will lead us to improve our manuscript.

We corrected these serious spelling errors (“hospitalized”, “hospitalization”, “generalize”) following your comment.